# Recent Advances in Poly(α-*L*-glutamic acid)-Based Nanomaterials for Drug Delivery

**DOI:** 10.3390/biom12050636

**Published:** 2022-04-25

**Authors:** Yu Zhang, Wenliang Song, Yiming Lu, Yixin Xu, Changping Wang, Deng-Guang Yu, Il Kim

**Affiliations:** 1School of Pharmacy, Shanghai University of Medicine & Health Sciences, Shanghai 201318, China; zhangy_21@sumhs.edu.cn (Y.Z.); b21070102030@stu.sumhs.edu.cn (Y.L.); xuyx@sumhs.edu.cn (Y.X.); 2Department of Polymer Science and Engineering, Pusan National University, Busan 46241, Korea; wenliang@usst.edu.cn; 3School of Materials Science & Engineering, University of Shanghai for Science and Technology, Shanghai 200093, China; 1935040425@st.usst.edu.cn

**Keywords:** cancer therapy, drug delivery system, nanomaterials, prodrugs, poly(*α*-*L*-glutamic acid)

## Abstract

Poly(*α*-*L*-glutamic acid) (PGA) is a class of synthetic polypeptides composed of the monomeric unit *α*-*L*-glutamic acid. Owing to their biocompatibility, biodegradability, and non-immunogenicity, PGA-based nanomaterials have been elaborately designed for drug delivery systems. Relevant studies including the latest research results on PGA-based nanomaterials for drug delivery have been discussed in this work. The following related topics are summarized as: (1) a brief description of the synthetic strategies of PGAs; (2) an elaborated presentation of the evolving applications of PGA in the areas of drug delivery, including the rational design, precise fabrication, and biological evaluation; (3) a profound discussion on the further development of PGA-based nanomaterials in drug delivery. In summary, the unique structures and superior properties enables PGA-based nanomaterials to represent as an enormous potential in biomaterials-related drug delivery areas.

## 1. Introduction

Cancer is one of the leading causes of death in humans. Although chemotherapy has gained enormous achievements in cancer treatment, the non-specific drug distribution-induced systemic side effects of anticancer drugs are still a Gordian knot [1,2]. Unremitting efforts have been made in the development of novel therapeutic regimens for a satisfactory curative effect [3,4,5,6,7]. Nanomaterials based on the incorporation of small molecule cancer drugs into the biocompatible polymers have attracted considerable attention because they can precisely deliver drugs to sites of action [8,9,10,11].

Poly(*α*-*L*-glutamic acid) (PGA) is a kind of synthetic polypeptide containing the monomeric unit *α*-*L*-glutamic acid [12]. Owing to their inherent properties including biocompatibility, biodegradability, non-immunogenicity, and till date, PGA-based nanomaterials have been extensively applied in biomedical fields such as cancer therapy, wound healing, medical devices, bio-sensing, and tissue regeneration [13,14,15]. Traditionally, PGAs are chemically synthesized by the ring-opening polymerization (ROP) of the *γ*-protected *L*-glutamate *N*-carboxyanhydrides (LG NCAs) [16,17]. After post deprotection, the bare carboxyl groups in each repeat unit of PGAs provide the high functionality for the chemical conjugation of molecules [18]. On the other hand, the water-soluble PGA moieties can serve as the hydrophilic building blocks of amphiphilic polymeric nanocarriers to physically entrap the therapeutic agents [19]. The pK_a_ of the pendent carboxyl side chains of PGAs is around 4.5. PGAs are negatively charged under physiological conditions; however, the pendent carboxyl side chains are positively charged when subject to acidic microenvironments, such as extracellular tumors (pH 6.8) and endosomes (pH 5.5) [12,20]. In addition, the ionic alternation transforms PGA moieties from hydrophilic and random-coiled conformation into hydrophobic and *α*-helical conformation, facilitating the stimuli-release of preloading.

In this review, we highlight the recently vital achievements in the development of PGA-based nanomaterials for drug delivery (Figure 1). A brief description of the synthetic strategies of PGAs will be firstly summarized. Then, the evolving applications of PGA in drug delivery systems (DDS), including the rational design, precise fabrication, and biological evaluation, will be extensively discussed. The current challenges and future perspectives of PGA-based nanomaterials are also presented.

## 2. Synthesis of PGA

Synthetic polypeptides are generally prepared from three methods: solid-phase peptide synthesis (SPPS), NCA polymerization, and microbial fermentation [12,20]. SPPS enables the automated preparation of the defined sequenced polypeptides via convenient isolation and purifications steps but is restricted to low molecular weight polymers (typically lower than 50 m) [21,22]. Microbial fermentation is an effective protocol to prepare high molecular weight and sequenced polypeptides, yet specialized and sophisticated equipment is not feasible for most synthetic laboratories [23,24,25]. NCA polymerization has been recognized as an economical and expedient technique that allows the fabrication of polypeptides with high yields, predetermined composition, narrow molecular weight distribution, and tuned functionality [26,27]. Furthermore, it’s a versatile and scalable method to synthesize polypeptides and is suitable for diverse types of NCAs. The past ten years has seen considerable developments in the polymerization of NCAs. This section will highlight the important advances on the chemical synthesis of PGAs.

Chemical synthesis of PGAs involves ROP of LG NCAs. Benzyl and tert-butyl are the most frequently used groups to protect the *γ*-carboxylic acid of *L*-glutamic acid because they are readily removable [28,29]. ROP of NCAs is generally triggered by the protic/nonprotic nucleophile or strong/weak base. The amine-initiated ROP of LG NCAs is the most prevalently used method to synthesis PGAs. There are two widely accepted mechanisms, the “normal amine mechanism” (NAM) and the “activated monomer mechanism” (AMM), which occur simultaneously and complicate the polymerization process (Figure 1) [30,31]. NAM typically initiates by a primary amine and presents a slow but controlled polymerization, whereas AMM generally mediates by a tertiary amine and gives a fast but uncontrolled polymerization.

Recent decades have witnessed great strides in the polymerization of NCAs, and various new initiators and catalysts have been reported for mediating controlled NCA polymerization. Transition metal complexes pioneered by Deming have emerged as efficient initiators to obtain a narrow molecular weight distribution, yet the inevitable removal of the metallic residues have restricted the general application of metal initiators (Figure 2) [32,33]. Considering the drawbacks of NAM and AMM, Hadjichristidis et al. proposed a peculiar mechanism, the “accelerated amine mechanism by monomer activation” mechanism (AAMMA), based on an “alliance” of primary and secondary amine initiators, including triethylaminetriamine, hexamethyldiamine and *N*,*N*′-dimethyl-1,2-ethanediamine (Figure 3) [34,35]. Silazane derivatives provide a possibility for the controlled NCA polymerization. As reported by Lu, hexamethyldisilazane, trimethylsilyl (TMS) and their derivatives could polymerize *γ*-benzyl-*L*-glutamate NCAs (BLG NCAs) in a well-controlled manner (Figure 3) [36,37]. Moreover, the Cheng group recently developed an auto-accelerated polymerization of *α*-helical polypeptides based on the TMS protected amine groups, where the synergistic interaction of macrodipoles between neighboring *α*-helices of polypeptides remarkably accelerate the ROP of NCAs [38]. The cooperative interaction involving *α*-helical of polypeptides also has been extended to diamine-initiated ROP of NCAs [39]. It is well known that NCA polymerization is significantly sensitive to humid environments and must be carried out under an inert atmosphere. Surprisingly, Liu et al. demonstrated that lithium hexamethyldisilazide can offer an extremely rapid ROP of NCAs either in the glovebox or in open vessel conditions (Figure 3) [40].

Hydrogen bonding interaction is a potent weapon to achieve controllable NCA polymerization. Our lab firstly discovered that imidazolium hydrogen carbonate salts, the precatalysts of *N*-Heterocyclic carbenes (NHC), can hydrogen bond with primary amine initiators as well as ω-terminus of the propagated polypeptide chain, generating linear structured polypeptides, whereas cyclic polypeptides are obtained in the absence of the primary amine initiators (Figure 4) [41,42]. Compared to amine initiators, the less nucleophilic alcohols usually lead to a slow initiation and poorly controlled polypeptides. Continuous efforts have been devoted to break the ice and enhance the nucleophilicity of alcohol. Based on the massive experiments and rigorous reasoning, it was reported that hydrogen bonding interactions with diazabicycloundecene, triazabycyclodecene, thioureas, and fluorinated alcohols can synchronously improve the nucleophilicity of the alcohol initiators, propagate hydroxyl terminus and activate the NCA monomers (Figure 5) [43,44,45].

Emulsification offers an alternative to traditionally amine initiated-NCA polymerization [46]. More recently, Cheng et al. have reported a water-in-oil emulsion to polymerize BLG NCAs by using a macroinitiator, affording an accelerating polymerization rate and the well-controlled polypeptide extension [47]. The concepts of frustrated Lewis pairs and self-assembly have also been explored to control the ROP of NCAs [48,49]. Aimed to achieve the controlled NCA polymerization, reaction conditions such as vacuum, temperature, photo, nitrogen flow, and solvent have also been carefully designed and optimized, except in developing emerging initiators and catalysts [50,51,52]. Given the impressive achievement in the preparation of polypeptides, diverse polypeptides-derived nanomaterials with predictable molecular weights, low dispersity, and defined functionalities can be constructed. Nevertheless, there is a huge gap between the synthetic polypeptides-derived nanomaterials and the clinically used one. In addition, the property and function of synthetic polypeptides are still far from matching the natural proteins. Further efforts in chemical synthesis of polypeptides with controllable monomer sequence, anticipated structure, and protein-like functions are required to achieve the preparation of clinically feasible biomaterials.

## 3. PGA-Based Nanomaterials for Drug Delivery

Chemotherapy is one of the crucial tumor therapeutic regimens in clinic [53,54,55,56]. To improve therapeutic efficacy and minimize nonspecific toxicity, the development of DDS that could enable the controllable release of chemical drug, alleviate premature drug leakage, and target specific tissues remains an enormous challenge [57,58,59,60]. PGA-based nanomaterials have gained increasing popularity in DDS due to their defined structure, tuned functionality, superior biocompatibility, and low immuneogenicity [61,62,63,64]. It is reported that PGA-based nanocarriers can specifically adhere to the *β*-glutamyl transpeptidase at the tumor cell membranes and overcome the serum inhibitory effect [65,66,67]. Various of chemotherapeutic drugs, such as doxorubicin (DOX), camptothecin (CPT), vascular disrupting agents (VDA), and platinum (Pt) drugs, have been chemically conjugated or physically encapsulated to PGA-based nanomaterials (Table 1) [68,69,70,71,72,73,74,75,76,77,78,79,80,81,82,83,84,85,86,87].

### 3.1. PGA-Based Nanomaterials as DOX Delivery Systems

DOX, an anthracycline antibiotic, is the most prevalently used chemotherapeutic drug for the treatment of various cancers, such as cancers of the breast, stomach, lung, thyroid, ovary, and bladder [88,89,90]. DOX can penetrate the endonuclear DNA and suppress DNA replication, leading to cell apoptosis. Nevertheless, the short blood circulation period and inevitable adverse effects including cardiotoxicity, nephrotoxicity, and myelosuppression significantly reduce its therapeutic outcomes [91,92]. Chemical conjugation and physical encapsulation of DOX to biodegradable PGA present a viable method to enhance its bioavailability and reduce side effects [93,94,95,96,97,98]. Traditionally, hydrophobic drug loading procedures contain the dissolution of the amphiphilic polymers and small molecular drugs in organic solvents, and the subsequent removal of the organic solvents by solvent evaporation or dialysis [99]. Unlike conventional encapsulation procedures, PGA-based anionic polymers are specifically complex, with cationic DOX through electrostatic interactions, omitting the use of detrimental organic solvents and achieving approximately 100% loading efficacy [100]. Chen et al. designed a poly(ethylene glycol)-*b*-poly(*α*-*L*-glutamic acid) (mPEG-*b*-PGA) nanocarrier entrapped with DOX through electrostatic interaction and an intermolecular hydrophobic stack (Figure 2A) [71]. The PEG segments were mainly oriented at the outer periphery of nanocarriers preventing the adsorption of protein and identification by the phagocyte system, whereas the PGA segments were primarily situated in the aqueous interior of nanocarriers warranting high loading of DOX. Cellular uptake tests suggested that the resultant nanocarriers were up-taken into A549 cells via endocytosis. Subsequently, the endosomal acidic condition triggered the destabilization of nanocarriers, resulting in the release of DOX to cytoplasm. Because of the enhanced permeability and retention (EPR) effect, the resultant ionomer complex exhibited a prolonged blood circulation period, decreased systemic toxicity, and enhanced therapeutic efficacy in the treatment of nonsmall cell lung cancer. To enhance the stability of PGA-based nanocarriers, hydrophobic units such as leucine (Leu) and phenylalanine (Phe) are incorporated to construct three monomeric units of the copolypeptides [100,101].

Pioneered by Kataoka, who firstly conjugated DOX to the pendent carboxyl acids of poly(ethylene glycol)-*b*-poly(*L*-aspartate) (mPEG-*b*-PLA), chemical conjugation of DOX onto polypeptides also have attracted considerable attention in cancer therapy [102,103,104,105,106]. Xiao et al. developed a pH and redox dual-stimuli poly(ethylene glycol)-*b*-poly(*γ*-propargyl-*L*-glutamate) (mPEG-*b*-PPLG) prodrug nanogel by simultaneously coordinating DOX through an acid-labile hydrazone bond and cross-linking with a redox sensitive 2-azidoethyl disulfide bond via one-step “click chemistry” [72]. The resultant mPEG-*b*-PPLG prodrug nanogels exhibited elevated stability during blood circulation and stimuli release of DOX in tumor cells. Recently, Vicent also developed a family of PGA-based combination conjugates bearing chemotherapeutic drug (DOX) and aromatase inhibitors (aminoglutethimide, AGM) for the treatment of breast cancer (Figure 2B) [73,107]. DOX was directly bound to the carboxyl groups of PGA either by amide bond or acid-labile hydrazone bond, whereas AGM was incorporated into PGA via a library of glycine (Gly) linkages (such as Gly linker, Gly–Gly linker, and Gly-Phe-Leu-Gly linker), which are readily cleaved by protease Cathepsin B. The controllable release of DOX and AGM in intracellular microenvironments enabled the superior therapeutic effects on primary tumor growth, apoptosis of cancer cells, and lung metastasis. PGA-based biomaterials provide a superior nanoplatform for small molecular drugs and achieved an enhanced therapeutic effect in tumor therapy. These pioneering examples pave the way for chemical conjugation and physical encapsulation of chemotherapeutics by using PGA-based nanomaterials.

### 3.2. PGA-Based Nanomaterials as Pt Drugs Delivery Systems

Hydrophobic Pt drugs like cisplatin (CDDP), carboplatin, and oxaliplatin have become promising candidates for the treatment of malignant tumors [108,109,110]. Pt drugs can contact DNA to disrupt its replication and eventually result in the apoptosis of tumor cells [111], whereas extremely low solubility and severe side effects significantly reduce its tumor therapeutic efficacy [112,113]. To overcome this restriction, a variety of polymeric nanocarriers have been explored to entrap Pt drugs [114,115]. Kataoka et al. firstly attempted to conjugate CDDP to the pendent carboxyl groups of mPEG-*b*-PLA [116]. To note, a series of Pt drugs coordinated mPEG-*b*-PGA, such as NC-6004 and NC-4016, have been assessed in phase III clinical trials for patients with advanced or metastatic pancreatic cancer [117,118]. The hydrophilic shell endows both NC-6004 and NC-4016 with a long blood circulation period and increased drug accumulation in the targeted tumor tissues through EPR effect.

Even so, the resistance and internalization dilemmas like free cisplatin still exist. A major reason lies in the steric repulsion of the dense PEG shell, which inevitably results in the PEGylated Pt drugs-conjugated nanocarriers bypassing the tumor tissues or failing to be phagocytosed by the tumor tissues [119]. To overcome these obstacles, chemical and physical dePEGylation induced by particular tumor microenvironments, such as pH, redox, and enzyme, have been deeply explored [120,121]. More recently, Xu et al. fabricated two types of poly(*L*-glutamic acid)-cisplatin (PGA-Pt) nanocarriers with cleavable PEG, which are sensitive to extracellular pH (pH_e_) and matrix metalloproteinases-2/9 (MMP-2/9) [74]. As displayed in Figure 3A, the pH_e_-sensitive 2-propionic-3-methylmaleic anhydride (CDM)-derived amide linkage and MMP-2/9-responsive cleaved peptide PLGLAG were designed to bridge PGA and PEG, generating pH_e_-sensitive PEG-pH_e_-PGA and MMP-2/9-sensitive PEG-MMP-PGA. CDDP was coordinated with the corresponding graft copolymers, yielding the polymer–metal complexed nanoplatforms, PEG-pH_e_-PGA-Pt and PEG-MMP-PGA-Pt. Cellular uptake assays revealed that PEG-PGA-Pt exhibited the limited cell internalization in SKOV3 cells due to the steric repulsion between the dense PEG shell and cell membrane. Conversely, PEG-pH_e_-PGA-Pt exhibited a significantly higher cell internalization in SKOV3 cells due to the dePEGylation triggered by the cleavage of the CDM-derived amide bond. The endosomal pH condition induced the instability of the bare PGA-Pt core, leading to the increased release of CDDP into cytol. Compared to the traditional PEG-PGA-Pt, the detachable PEG-pH_e_-PGA-Pt and PEG-MMP-PGA-Pt not only retained the prolonged circulation time, the pH and MMP detachable PEGylated PGA-Pt nanoformulations enabled the enhanced cell internalization toward the high-grade serous ovarian cancer, eventually leading to the up-regulated antitumor efficacy.

Tang et al. integrated the merits of the “receptor-mediated cellular uptake” and “multi-drug delivery” into one nanoformulation (Figure 3B) [75]. Docetaxel (DTX) and CDDP were co-encapsulated into the amphiphilic poly(*L*-glutamic acid)-*g*-*α*-tocopherol/polyethylene glycol (PGA-*g*-Ve/PEG) nanocarriers through hydrophobic and chelation interaction, followed by the periphery decoration of an avb3 integrin targeting peptide c(RGDfK). Thanks to the targeting c(RGDfK), DTX/CDDP co-encapsulated nanoformulation exhibited a synergistically increased accumulation rate and retention time in mouse melanoma cells. Folic acid (FA), an active targeting ligand, has also been extensively utilized for targeted CDDP delivery. Qiao et al. recently designed the CDDP-loaded maleimide-poly(ethylene oxide)_114_-*b*-poly(*L*-glutamic acid)_12_ (Mal-PEG_114_-*b*-PLG_12_) vesicles for the targeted delivery of CDDP to tumor sites [76]. CDDP complexed to PGA moieties induced the self-assembly of the copolymer into vesicular morphologies via the formation of a hydrophobic domain, while PEG blocks served as the corona and interior layer of the vesicular morphologies. The reactive maleimide groups on the vesicle periphery could conjugate with FA thiol, yielding an active targeted DDS, which presented distinctly high cellular uptake and desired cytotoxicity toward HeLa cells. Targeting agents enable the targeted CDDP delivery to tumor sites, yet the dedicated and complicated modification procedures also increase the potential system toxicity.

### 3.3. PGA-Based Nanomaterials as CPT Delivery Systems

As a topoisomerase I inhibitor, CPT which is derived from the Chinese tree Camptotheca acuminata, can induce a variety of tumor cell apoptosis [122,123,124,125]. Unfortunately, the relatively low aqueous solubility and pH-dependent lactone ring stability of CPT severely constrain its clinical application [126]. Towards this end, both Singer and Klein’s groups demonstrated that the water solubility and lactone ring stability could greatly be enhanced by the conjugation of CPT to the residing carboxylic acid of PGA [127,128]. Researchers have also combined CPT with other chemotherapeutic drugs for synergistic cancer treatment [77,129]. Xiao et al. developed a redox responsive nanoformulation via the self-assembly of poly(*L*-glutamic acid)-*g*-poly(ethylene oxide) (PGA-*g*-mPEG) based CPT conjugate and simultaneous entrapment of DOX by hydrophobic interaction (Figure 4) [77]. CPT was linked to PGA-*g*-mPEG via a disulfide bridge which was readily detachable in a glutathione (GSH) environment. It was observed that the intracellular GSH concentration plays a decisive role in the release of DOX and CPT from nanoformulation. As proved by the flow cytometry and the cellular uptake tests, the acidic endo-lysosomal microenvironment induced the release of DOX, GSH in the cytoplasm, and cleaved the disulfides, releasing CPT from the resultant nanoformulation. The low combination index value (approximately ~0.3) substantiated the valid cancer cell apoptosis based on the nanoformulation co-loaded with CPT and DOX. A semisynthetic analog of CPT, 7-Ethyl-10-hydroxy camptothecin (SN38), has been approved by FDA for colorectal carcinoma therapy [130]. Recently, Tamaddon et al. prepared a double hydrophilic poly(2-ethyl 2-oxazoline) block poly (*L*-glutamic acid) (PEtOx-*b*-PGA) prodrug by coupling SN38 to the pendent carboxyl group of PGA [78]. Compared to free drugs, cell culture assays displayed a higher intracellular accumulation and at least four times more specific cytotoxicity than the coupled SN38 in the CT26 cell line. Moreover, the as-prepared SN38 conjugate exhibited outstanding anti-tumor activity and was significantly superior to commercial irinotecan, especially on advanced tumors with a reduced mortality rate of 2.5 times. All these studies suggest that CPT-derived topoisomerase I inhibitors can exert their tumor cell apoptosis effect in tumor therapy. Further efforts are needed to investigate the more detailed action mechanism of these nanoformulations.

### 3.4. PGA-Based Nanomaterials as VDA Delivery Systems

VDA can selectively modulate tumor vasculature and rapidly induce the shutdown of tumor blood vessels, leading to widespread tumor cell ischemic necrosis [131,132]. Combretastatin A4 (CA4) is a crucial agent for clinical cancer therapy. As a kind of microtubule depolymerizing agent, CA4 can attach to the colchicine adhering site of *β*-tubulin, resulting in cytoskeletal destabilization and morphological variation of the endothelial cell [133,134,135]. The poor aqueous solubility of CA4 is the greatest hindrance for the extensively clinical application. Tong et al. designed a polymeric CA4 conjugate via coordination of CA4 to poly(*L*-glutamic acid)-CA4 (PGA-CA4) (Figure 5A) [79]. Intra-tumor distribution experiments indicated that PGA-CA4 nanoconjugates were predominantly localized around tumor blood vessels due to the active targeting property of CA4. This enabled the long-term release of CA4 inside solid tumor cells. The gradually increased CA4 concentration around tumor blood vessels caused the steady tumor blood deprivation and effective tumor regression (Figure 5B). Owing to the vascular-dependent distribution character, the obtained PGA-CA4 exhibited a long retention time in plasma and the murine colon C26 tumor cell compared to commercial combretastatin-A4 phosphate (CA4P). After a single administration, PGA-CA4 induced enduring angiorrhexis and tumor destruction in 72 h, leading to a tumor suppression rate of 74%.

However, several side effects, such as polarization induced by PGA-CA4, significantly restricted the antitumor activity. To this end, the same group also combined this nanomedicine with other antineoplastic agents for enhanced cancer therapy [136,137]. For instance, they utilized the phosphoinositide 3-kinase gamma isoform (PI3K*γ*) selective inhibitors synergizing with PGA-CA4 to reduce the immunosuppressive effects (Figure 6) [80]. The number of M2-like tumor-related macrophage obviously decreased while the cytotoxic T lymphocytes markedly improved due to PI3K*γ*inhibitor. Remarkably, the combination of PI3K*γ* inhibitor and PGA-CA4 prevented the tumor growth and extended the mean survival time, significantly enhancing the tumor therapeutic efficacy (Figure 6). Even though CA4-conjugated nanocarriers effectively inhibited tumor growth and tumor proliferation, CA4 could not be tested by multispectral optoacoustic tomography and immunofluorescence assay, hindering the observation of the intra-tumor distribution of CA4.

### 3.5. PGA-Based Nanomaterials as Gas Molecule Delivery Systems

Mammalian tissues generate many kinds of gas molecules, such as nitric oxide (NO), carbon monoxide (CO), and sulfur dioxide (SO_2_), which play a transmitter role in a series of biological activities and regulate the biochemical or physiological processes in the human body [138]. Recently, gas therapy has become an emerging tumor therapeutic technique because there is no drug resistance, it has minimal side effects and there is no byproduct [139]. Numerous gas nanogenerators are designed to delivery and produce safe gas molecules for the treatment of tumors [140,141]. NO, is an endogenously generated radical gas molecule, involved in various physiological functions, such as cardiovascular homeostasis, neurotransmission, and immune response to infection and angiogenesis [142]. NO can modulate P-glycoprotein expression without multi-drug resistance at low dosages, while high concentrations of NO can damage DNA and mitochondria in solid tumors, leading to cell mortality [143]. Hong et al. fabricated (poly-*L*-lysine/poly-*L*-glutamic acid)_n_ (PLL/PGA)_n_ multilayer films with different thicknesses for controlled NO releasing [81]. By applying the layer-by-layer self-assembly approach, PLL and PGA were employed to construct the multilayer films via electrostatic interaction, where PLL served as the positively charged blocks and PGA acted as the negatively charged blocks. A proton-responsive NO donor, *N*-diazeniumdiolate, was loaded into (PLL/PGA)_n_ multilayer films via a high pressure reaction under NO atmosphere. The as-obtained (PLL/PGA)_n_ multilayer films displayed a continued NO releasing behavior, suggesting the controllable NO delivery for tumor treatment. SO_2_ has been recognized as a promising gasotransmitter for regulation of the cardiovascular system. Xiao et al. developed a polymeric GSH-responsive nanomedicine of SO_2_ to combat MCF-7 ADR human breast cancer cells in synergy with DOX [82,144]. *N*-(3-azidopropyl)-2,4-dinitrobenzenesulfonamide (AP-DNs), a small molecular generator of SO_2_, was coupled onto the pendent groups of methoxy poly(ethylene glycol)-*block*-poly (*g*-propargyl-*L*-glutamate) (mPEG-*b*-PPLG) copolymer via “click chemistry”, yielding the polymeric nanomedicine of SO_2_, mPEG-*b*-PLG (DNs). DOX was finally encapsulated into mPEG-*b*-PLG (DNs) nanomedicine via self-assembly. Upon GSH triggering, the obtained mPEG-*b*-PLG (DNs) nanomedicine simultaneously released SO_2_ and DOX, causing an enhancement of reactive oxygen species (ROS) in tumor tissue and synergistic anti-proliferation effects against MCF-7/ADR cells.

### 3.6. PGA-Based Nanomaterials as Co-Delivery Systems

Co-delivery of dual antineoplastic agents in a polymeric nanocarrier has attracted enormous interest due to the synergistic therapeutic effect [145,146,147,148,149]. Tremendous efforts have been devoted to exploring the combined treatment based on chemotherapy, phototherapy, biological therapy, and radiation therapy. Photodynamic therapy (PDT) is a kind of phototherapy involving light irradiation and photosensitizer. Upon being activated by specific light irradiation, photosensitizer can produce ROS to induce the tumor cell apoptosis [150]. Yu et al. designed CA4 and porphyrin (5, 10, 15, 20-tetraphenylporphyrin, TPP)-conjugated nanomedicines, CA4-conjugated poly(*L*-glutamic acid)-*graft*-methoxy poly(ethylene glycol) (PGA-*g*-mPEG-CA4), and TPP-conjugated PGA-*g*-mPEG (PGA-*g*-mPEG-TPP), for combined vascular disrupting photodynamic therapy [83,151]. Upon laser irradiation, the PGA-*g*-mPEG-CA4 nanomedicines exhibited superior antitumor ability and PGA-*g*-mPEG-TPP nanomedicines generated dioxygen to kill cancer cells. However, the combination of PGA-*g*-mPEG-CA4 and PGA-*g*-mPEG-TPP nanomedicines connected the effect of vascular blockage and photodynamic cell apoptosis, attaining the efficacy of interior and exterior tumor cell killing.

Photothermal therapy (PTT), another type of phototherapy, can result in apoptosis or necrocytosis of the tumor issues and inhibit tumor proliferation by inducing the partial hyperthermia effect of photothermal agents [152]. Li et al. constructed gold nanorods (GNRs) with CDDP-methoxy poly(ethylene glycol)-*graft*-poly(*L*-glutamic acid) (CDDP-mPEG-*g*-PGA) wrapping and FA decoration (FA-GNR@Pt) for the targeting of chemo-photothermal therapy of breast cancer (Figure 7A) [84]. The chemical conjugation of GNRs to mPEG-*g*-PGA copolymers with thiol groups could efficiently remove the cetyl trimethylammonium bromide moieties and minimize the toxicity of GNRs. To avoid protein absorption and prolong blood circulation, CDDP was complexed into the inner PGA core and FA were linked to the outer PEG corona. The as-prepared FA-GNR@Pt prodrug significantly suppress the growth and lung metastasis of the 4T1 breast tumor due to FA- mediated tumor targeting effects, and CDDP caused cellular apoptosis in synergy with near infrared laser illumination-induced cellular necrosis and ablation of the peripheral blood vessels.

Owing to easy operation and deep penetration into soft tissues, magnetic resonance imaging (MRI)-based techniques have been extensively applied in clinic treatment. Conventional T1-type contrast agents are usually restricted because of the risk of accumulated toxicity because of poor sensitivity. Du et al. proposed a noncytotoxic targeting polymer vesicle based on FA or diethylenetriaminepentacetatic acid (DTPA) poly(*L*-glutamic acid)-*b*-poly(*ε*-caprolactone) (FA/DTPA-PGA-*b*-PCL) (Figure 7B) [85]. The DOX and Gd(III) co-encapsulated asymmetrical polymer vesicles were obtained via the self-assembly of FA-PGA-*b*-PCL copolymers, where the hydrophobic PCL blocks act as the vesicular membranes and the hydrophilic PGA blocks serve as the vesicular coronas. Among which, the longer PGA blocks linked with FA contributed as the external coronas, whereas the shorter PGA blocks attached with DTPA acted as the internal coronas. Compared to traditional DTPA-Gd, such asymmetrical vesicles presented a high T1 relaxivity of 42.93 mM^−1^·s^−1^ and drug loading efficiency of 52.6% for DOX·HCl (Figure 7C,D). Moreover, in vivo MR imaging assays suggested an evident enhancement of the signal intensity around the targeted tumor sites. Compared to chemotherapeutics-based monotherapy, these dual therapy systems significantly improved the therapeutic effect in a synergistic way. New approaches with more simple, efficient, and economical properties are desired to fabricate the dual therapeutic nanoplatforms.

### 3.7. PGA-Based Nanomaterials as Protein Delivery Systems

Since the approval of the first protein drug Humulin^®^ in 1982, hundreds of therapeutic protein drugs have been approved [153,154]. Peptide-based therapeutics have gained great momentum, owing to their high specificity and efficiency, fewer side effects, and that they are tolerated well by the human body [155,156]. Unfortunately, the peculiar hierarchical architectures of peptide drugs also endow them with inherent pharmaceutical defects, including poor stability, immunogenicity, and shorter retention time [157,158]. To overcome these obstacles, therapeutic regimens are challenged to modify the molecular structure and formulation, such as the covalent incorporation of PEG and glycolic acid, or the construction of polymeric nanocomposites [159,160]. Kataoka et al. designed a (1,2-diaminocyclohexane)platinum(II) (DACHPt)-conjugating poly(ethylene glycol)-*b*-poly-(*L*-glutamic acid) (PEG-*b*-PGA) polymeric micelle with cyclic Arg-Gly-Asp (cRGD) ligand molecules for the targeted delivery of platinum therapeutic prodrugs to glioblastomas [86]. Compared to the polymeric micelle bearing nontargeted ligand (cyclic-Arg-Ala-Asp), cRGD-conjugated nanocarriers (cRGD/m) accumulated, accelerated, and possessed superior permeability from vessels into the tumor parenchyma (Figure 8A). The rapid accumulation of cRGD/m into tumor cells through an active internalization route should be responsible for the significantly improved tumor therapeutic effect in the treatment of U87MG glioblastoma.

Recently, Tang et al. reported a self-amplifying, therapeutic tumor-homing nanoplatform (A15-PGA-CA4) that works via a chain reaction mechanism [87]. The blood coagulation-targeting peptide (GNQEQVSPLTLLKXC, termed A15) was conjugated to poly(*L*-glutamic acid)-*graft*-maleimide poly(ethylene glycol)/combretastatin A4 (PGA-*g*-PEG-Mal/CA4) by thiol–maleimide “click chemistry” (Figure 8B). After intravenous injection, A15-PGA-CA4 released CA4 and initiated the chain reaction cycles: (1) intratumoral hemorrhage: CA4 selectively disrupted the established tumor blood vessels, leading to hemorrhage within treated tumor cells; (2) target blood coagulation factor XIIIa (FXIIIa) amplification: the hemorrhage triggered an intratumoral coagulation cascade effect, in which FXIII was activated into FXIIIa, causing targeted amplification; (3) blood clot binding: FXIIIa guided A15-PGA-CA4 in the blood stream to tumor tissues via binding to blood clot; (4) CA4 release in tumor sites: A15-PGA-CA4 continuously released CA4 in cancer cells and the next cycle started subsequently. A15- PGA-CA4 enhanced the content of the targeted FXIIIa via chain reaction. Owing to the superb targeting potency of A15- PGA-CA4, the tumor therapeutic effect against large C26 tumors was significantly improved. These impressive examples reaffirm the high specificity and efficacy of protein drugs which can be well exerted in the preparation of tumor-targeting biomaterials. Moreover, the exact chain reaction mechanism of A15- PGA-CA4 guided the combined cancer therapy based on co-delivery of chemical drugs and proteins.

### 3.8. Other PGA-Based Hybrids as Drug Delivery Nanovehicles

Owing to their intrinsic biodegradable and biocompatible characteristics, PGA has also been designed as a diverse kind of nanocomposite for biomedical applications [161,162,163,164,165]. For instance, Cheng et al. developed a tumor-targeting siRNA delivery nanoplatform based on a *α*-helical cell-penetrating polypeptide (PVBLG-8) and a random-coiled PGA [166]. The cationic and rigid PVBLG-8 exhibited weak siRNA condensation ability, whereas the anionic and flexible PGA acted as a stabilizer to entrap the siRNA within the molecular entanglement between PVBLG-8 and PGA. During circulation, the as-obtained nanonetwork displayed an anticipated serum stability and significantly increased tumor accumulation via the EPR effect. Vicent et al. designed a pH-responsive biodegradable polypeptide-corticosteroid conjugate (PGA-FLUO@HA-CP) for the topical treatment of psoriasis [167]. The PGA conjugation of fluocinolone acetonide (FLUO) contributed to penetration and drug exposure, thus improving targeting to the viable epidermis. Remarkably, the encapsulation of PGA-FLUO within the hyaluronic acid-poly(*L*-glutamate) cross polymer (HA-CP) vehicle further enhanced this targeting effect by boosting skin permeation. PGA-FLUO in synergy with HA-CP precisely delivered FLUO to the inflammatory skin layers, avoiding the potential side effects. Recently, Du et al. successfully prepared the bone-targeting polymeric vesicles for the effective treatment of postmenopausal osteoporosis [168]. These polymeric vesicles were self-assembled from poly(*ε*-caprolactone)_28_-*block*-poly[(*L*-glutamic acid)_7_-*stat*-(*L*-glutamic acid-alendronic acid)_4_] (PCL_28_-*b*-P[Glu_7_-*stat*-(Glu-ADA)_4_]) copolymers. The hydrophilic P[Glu_7_-*stat*-(Glu-ADA)_4_] chains acted as the coronas while the hydrophobic PCL chains served as the membranes which could encapsulate *β*-estradiol (E2) inside the vesicles. The conjugated ADA on the coronas enabled the vesicles with superior bone affinity and acted synergistically with E2 to increase bone mass and thus reached an enhanced osteoporosis treatment effect.

## 4. Conclusions and Perspectives

In this work, we have summarized recent vital advances of PGA-based nanomaterials in drug delivery. The inherent biodegradable, biocompatible, and ionic charging properties make PGA an attractive candidate for the construction of biomedical polymeric materials. The pensile carboxyl side groups of PGA provide abundant attachment sites for therapeutic drugs such as DOX, CPT, and Pt drugs, through either chemical incorporation or physical entrapment. PGA-based conjugates with defined structures and specific functions also have been prepared to deliver gas molecules or protein drugs. Owing to the synergistic therapeutic effects, diverse therapeutic techniques combined with chemotherapy have been widely reported by researchers. Moreover, the ionic-charged PGA have also been utilized in a variety of biomedical applications, such as antimicrobial complexes, vaccine adjuvants, tissue regeneration, and medical devices [169,170,171,172]. For example, clinical complications induced by tissue adhesion during surgery usually increase the degree of pain for patients [173,174]. Recently, Ko et al. designed a PGA-based anti-adhesion membrane conjugated with the anti-inflammatory drug ibuprofen for preventing tissue adhesion [175]. Owing to the proper hydrophilicity of PGA, this membrane exhibited superior wound coverage without the contraction. This study will certainly guide the future design and clinical applications of PGA-based nanomaterials.

Great efforts are required to achieve the translation of PGA-based nanomaterials from laboratory to clinic. PGA-based nanomaterials are confronted with two consistent challenges—enhanced therapeutic efficacy and scalable synthesized protocol. In addition to high stability during the systemic circulation, PGA-based nanomaterials should release the payload in the tumor sites. In consideration of the pendent carboxyl groups which endow PGA with pH-responsive property, various mono- or multi-stimuli responsive PGA-based nanomaterials have been proposed. For example, acid-labile hydrazone bonds and reduction-sensitive 2-azidoethyl disulfide linkage were utilized to construct dual-stimuli PGA-based prodrug with intracellular pH and redox responsiveness [91]. Furthermore, photo- and enzyme-responsive PGA conjugates also have been extensively reported, whereas further efforts are pressingly required to understand the mechanism of intratumor/intracellular release of payload [148,176]. High drug loading capacity and on-demand drug releasing ability at anticipated sites also play a crucial role in enhancing therapeutic efficiency. With the thrilling advances in NCA polymerization, synthetic polypeptides with well-defined structures and functionalities afford the novel building blocks for polymeric biomaterials. However, the property and function of these synthetic polypeptides are still far from matching the natural proteins. Therefore, it is a good opportunity to synthesize more functionalized polypeptides with predictable molecular weights, tunable monomer sequences, and controllable termini. On the other hand, the ingenious design of PGA-based nanomaterials usually involves multiple syntheses and purification procedures, which represent a formidable challenge for scalable and reproducible preparation of materials.

## Data Availability

Data supporting this publication are available from the corresponding authors.

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
