# Peer review of "Recent Advances in Poly(α-*L*-glutamic acid)-Based Nanomaterials for Drug Delivery"

_biomolecules, 2022, doi:10.3390/biom12050636_

Round 1
Reviewer 1 Report
The manuscript is " Recent advances in poly(α-L-glutamic acid)-based nanomateri-2 als for drug delivery ".
General comments:
This study summarized the PGA-based nanomaterials synthesis and drug delivery in biomedical applications. However, there is still some work to be done.
- More and detailed the schematic illustration of PGA synthesis.
- Again, be careful to confirm the abbreviation's description.
- It can increase the direction and type of FGA that can be applied to biomedical materials in the future
Author Response
We appreciate reviewer's valuable suggestions and we have carefully revised all points in our revised manuscript.

Reviewer 2 Report
In this manuscript entitled 'Recent advances in poly(α-L-glutamic acid)-based nanomaterials for drug delivery', they highlighted the recently vital achievements in the development of PGA-based nanomaterials for drug delivery. A brief description of the synthetic strategies of PGAs will be firstly summarized. Then, the evolving applications of PGA in drug delivery system (DDS), including the rational design, precise fabrication, and biological evaluation, will be extensively discussed. The current challenges and future perspectives of PGA-based nanomaterials are also presented. However, the manuscript needs further improvement to meet the publication requirements of this journal. Some questions and suggestions are as follow :
- The abstract of this manuscript is not logical enough.
- The process of precise drugs delivery to sites should be discussed.
- The mechanism of stimuli-release of preloading should be explained more clearly.
- The manuscript still lacks an adequate use of the published literature to introduce the background of this study. The following articles are relevant to your work which is suggested to be cited: 1038/s41565-021-00976-3, 10.1016/j.jobab.2020.04.003.
- The size and order of the outermost words in Figure 1 should be adjusted.
- The title of the section 3 ( PGA-based Nanomaterials for Cancer Therapy) is not specific enough and the content is not clear.
- The quality of the Figure 2B and Figure 7A, B needs to be improved. The figures of following articles are favorable which is suggested to be referred: 1016/j.jobab.2021.06.001.
- The comparison to cationic polymeric carriers is confused (line 134), because cationic polymeric carriers are not mentioned in the review.
- There are still many typos and format issues in the manuscript. Authors should carefully check again.
- Each section should include a summary and your point of view.
- The font of all pictures in the manuscript should be consistent and appropriate. The figures of following articles are favorable which is suggested to be referred: 1039/C9CS00839J, 10.1016/j.jobab.2020.10.001.
- The chemical reactions related to the synthesis of PGA should be summarized.
- The structure of the section 4 makes me very confused. If it is parallel to the section 3, the content is lacking.
- You should add a table to compare the advantages and disadvantages of the work summarized above. The following articles are relevant to your work which is suggested to be cited: 1016/j.jobab.2020.07.003, 10.1016/j.jobab.2020.04.003.
Author Response

(The authors gave the same response as above.)

Reviewer 3 Report
The review article provides well presented and useful information related with recent advances in PGA-based nanomaterials in drug delivery systems and can be accepted in present form already at first stage of submission though addition of some minor details can improve the quality of Introduction section.
More citations can be provided in section 3.1 PGA-based Nanomaterials as DOX Delivery Systems, for example Jaimes-Aguirre et al. Biodegradable poly(D,L-lactide-co-glycolide)/poly(L-γ-glutamic acid) nanoparticles conjugated to folic acid for targeted delivery of doxorubicin. Materials Science and Engineering C 76 (2017) 743–751.
More specific additional information could be beneficial in the section 5,
Author Response

(The authors gave the same response as above.)

Round 2
Reviewer 1 Report
The revised version is now acceptable.
Reviewer 2 Report
The authors answered very well the raised questions. The current version could be accepted for publication.